# Distributing Plant Developmental Regulatory Proteins via Plasmodesmata

**DOI:** 10.3390/plants13050684

**Published:** 2024-02-28

**Authors:** Joyce M. Schreiber, Erik Limpens, Jeroen de Keijzer

**Affiliations:** 1Laboratory of Cell and Developmental Biology, Wageningen University and Research, Droevendaalsesteeg 1, 6708 PB Wageningen, The Netherlands; joyce.schreiber@wur.nl; 2Laboratory of Molecular Biology, Wageningen University and Research, Droevendaalsesteeg 1, 6708 PB Wageningen, The Netherlands; erik.limpens@wur.nl

**Keywords:** plasmodesmata, intercellular communication, KNOTTED1, SHORTROOT, non-cell-autonomous transcription factors, plant meristem

## Abstract

During plant development, mobile proteins, including transcription factors, abundantly serve as messengers between cells to activate transcriptional signaling cascades in distal tissues. These proteins travel from cell to cell via nanoscopic tunnels in the cell wall known as plasmodesmata. Cellular control over this intercellular movement can occur at two likely interdependent levels. It involves regulation at the level of plasmodesmata density and structure as well as at the level of the cargo proteins that traverse these tunnels. In this review, we cover the dynamics of plasmodesmata formation and structure in a developmental context together with recent insights into the mechanisms that may control these aspects. Furthermore, we explore the processes involved in cargo-specific mechanisms that control the transport of proteins via plasmodesmata. Instead of a one-fits-all mechanism, a pluriform repertoire of mechanisms is encountered that controls the intercellular transport of proteins via plasmodesmata to control plant development.

## 1. Introduction

While the cell is often seen as a singular unit of life, multicellular organisms across the various kingdoms of life have evolved ways through which neighboring cells form physical interconnections [1,2]. For land plants, this interconnection is achieved via cell wall-traversing and tunneling structures called plasmodesmata. Plasmodesmata enable the transport of soluble micro- and macromolecules, thereby providing direct coupling between the cytosol of neighboring cells. This cytosolic continuity is referred to as the symplast, and hence plasmodesma-mediated molecular exchange is known as symplastic transport. Together with secretion to and uptake from extracellular spaces (known as apoplastic transport), it forms the principal way by which neighboring plant cells can exchange molecules.

In the basic structure of a plasmodesma, the cytosol-to-cytosol connective layer (often referred to as the ‘cytoplasmic sleeve’ or, in older literature, as the ‘cytoplasmic annulus’) is bounded on the outside by the plasma membrane and on the inside by a tightly appressed strand of endoplasmic reticulum (ER) called the desmotubule. While this overall structure also means that these membrane systems as well as the ER lumen have continuity from cell to cell, the transport of macromolecules occurs principally via the cytoplasmic sleeve [3]. The membrane configuration at plasmodesmata, alongside the local properties of the surrounding cell wall, is important in determining the space for intercellular fluxes through the cytoplasmic sleeve. Since the typical size of the cytoplasmic sleeve (on the order of 1–10 nm) is in the same range as that of cellular macromolecules, alterations in the sleeve’s effective aperture affect what macromolecular cargoes can and cannot traverse through the channel. This size-selective property of a plasmodesma (or, more typically, a population of plasmodesmata on a cellular interface) is called the size exclusion limit (SEL).

From a functional perspective, plasmodesmata form conduits not only for resource sharing but also for the exchange of signaling molecules. Signaling cascades operating across tissues via plasmodesmata occur in the context of normal development as well as when a plant mounts a response to environmental stimuli [4]. The molecules encoding such signals appear to span diverse types, ranging from plant hormones [5,6,7,8,9,10] to RNA species (reviewed in [11,12]) and proteins.

Plant development is primarily orchestrated by meristematic regions, and the activity and cellular identities in these meristems are informed by spatial signals [13]. Plasmodesma-mediated transport forms an important conduit for this. This is reflected by several developmental transitions being accompanied by active alterations to the plasmodesmal SEL. Interfering with the number of plasmodesmata or their SEL often disrupts developmental patterning (well-known examples include embryogenesis and lateral root formation [14,15]; reviewed in [4]). Thus, spatio-temporal control over developmental processes stems in part from the correct partitioning of molecules carrying developmental instructions via plasmodesmata. In this review, we focus especially on the transport of mobile proteins, most notably transcription factors, that control developmental decisions in distal tissues [16,17]. For convenience, we will refer to these proteins as plasmodesmal cargo proteins.

The plasmodesmal transport of such cargo proteins is the concerted outcome of regulation at two levels: (1) mechanisms determining overall plasmodesmal connectivity across a particular cell–cell interface and (2) mechanisms controlling the permissiveness of plasmodesmal transit of a particular protein. In this review, we will discuss both aspects and focus on the cellular and molecular processes underpinning the regulatory mechanisms on these two levels. First, we will discuss the biogenesis, form, and dynamics of plasmodesmata that set the first level of control (Section 2). This will be followed by an overview of the emergent molecular mechanisms that control the cell-to-cell movement of individual developmental regulatory proteins (Section 3). Finally, we provide a brief perspective on the integration of regulatory mechanisms and their future study (Section 4).

## 2. Setting the Stage for Intercellular Transport: Biogenesis and Regulation of Plasmodesmata

The formation and frequency of plasmodesmata sets the baseline potential for communication. On top of this, the diversification of plasmodesmal shape and dynamic regulation can serve to ultimately determine the effective potential for intercellular communication. In this section, we will describe these processes and the accompanying cellular and molecular mechanisms. The moment of inception of plasmodesmata can be either during cytokinesis or during interphase, distinguishing what are called primary and secondary plasmodesmata, and the associated mechanisms of their biogenesis are different [18,19].

### 2.1. Formation and Control over Primary Plasmodesmata

Primary plasmodesmata are formed within the nascent cell wall segment, which is generated during plant cytokinesis to divide the two daughter cells. This cell wall segment is generated via a radially expanding, membranous disc termed the cell plate [20]. The cell plate initiates and expands by the fusion of vesicles, after which membrane remodeling processes and polysaccharide deposition gradually shape it into a smooth partitioning segment that connects to the parental cell wall. In the earliest stages of cell plate formation, strands of ER are already intermingled among the vesicles [21,22,23]. The ER remains closely associated as the cell plate vesicles fuse into larger, sheet-like structures. This cell plate-associated ER is connected to the larger ER network [23]. While the ER may have a general role in fostering cell plate synthesis [24], the ER strands that traverse the cell plate are critical to the establishment of primary plasmodesmata [22] (Figure 1A). Such bridging strands of ER between the two daughter cells appear to resist the cell plate remodeling processes that close and smoothen the cell plate membrane compartment [24,25,26]. The strand of ER persists throughout cell plate maturation and becomes tightly appressed, ultimately forming the desmotubule proper [18,22] (Figure 1A). Thus, the interaction between the forming cell plate and traversing ER strands during cytokinesis ‘seed’ the plasmodesmata on the incipient cell–cell interface.

During the transition from ER entrapment in the cell plate to plasmodesma, stabilization and shaping of the ER are important processes. Several protein families that are able to facilitate these activities are discussed below.

Reticulons are membrane-embedded proteins that facilitate membrane curvature and constrict ER membranes by oligomerization [27,28]. Of the 16 conventional reticulons (RTNs) found in Arabidopsis, reticulons RTN3 and RTN6 were found to be particularly highly enriched at the cell plate-associated ER [29]. This, combined with indications of their sustained presence at the plasmodesmata (and their ability to interact with viral movement proteins) [30,31], makes reticulons likely factors in constricting the desmotubule (Figure 1A). However, their precise contribution to plasmodesmal biogenesis and function has not been gauged yet via, for instance, loss-of-function studies.

The intimate and sustained interaction between ER and cell-plate membranes during primary plasmodesmal biogenesis highlights a potentially important role for proteins capable of stably interconnecting the two membranes, an activity known as ‘tethering’. Indeed, using electron microscopy, bridging elements between the cell plate and ER membranes have been directly observed [24]. While membrane-tethering proteins are more widely known for stabilizing membrane–membrane contact prior to a fusion event [32], in the plasmodesmal context, tethers are typically discussed as mediators of membrane contact sites [33]. A membrane contact site is a defined area where two different (organellar) membranes are kept in close proximity and where molecular exchange can take place in the absence of membrane fusion (reviewed in [34]). Tethering proteins could, therefore, be prime molecular constituents to structure and stabilize a forming plasmodesma.

At fully formed plasmodesmata, an important class of plasma membrane and ER membrane tethering proteins is the family of Multiple C2-domains and Transmembrane Proteins (MCTPs) [35]. Their N-terminal C2-domains can interact with the inner leaflet of the plasma membrane, while the C-terminus traverses the ER membrane [35,36]. Recently, MCTPs were shown to accumulate and oligomerize early during plasmodesmal biogenesis, and in their absence, densities of primary plasmodesmata markedly dropped [26]. Additionally, the MCTP C-terminal domain inserted in the ER membrane was found to have ER-tubulizing properties, akin to reticulons [26]. Thus, MCTPs are tethering proteins with an aptitude to stabilize and constrict ER strands traversing the cell plate, thereby contributing to successful plasmodesma formation (Figure 1A). As discussed later (in Section 3) MTCPs are also important players in the transport of individual proteins across plasmodesmata.

Another family of tethering proteins with a similar molecular architecture to MCTPs are synaptotagmins (SYTs). Since SYTs additionally have a domain involved in the transfer of lipids among membranes in close proximity, their role might extend beyond acting as structural elements [37]. In Arabidopsis, SYTs are enriched at early plasmodesmata [38]. At later stages, SYTs are found only mildly enriched at plasmodesmata and reside broadly at ER–plasma membrane contact sites [38,39]. In SYT mutants, the ER tubule leading to the desmotubule becomes less constricted [39]. SYTs also interact with diverse viral movement proteins, and their presence enables viral movement protein and virion cell-to-cell transmission, supporting a role at or near plasmodesmata [39,40]. SYTs thus deserve further attention to establish their role in plasmodesmal biogenesis (Figure 1A).

Since plasmodesmal frequency is a major determinant of the degree of intercellular connectivity between two newly formed daughter cells, activities such as ER-membrane tubulation, stabilization, and abscission, that operate during cytokinesis, are likely important in setting consequent tissue connectivity. An indication that plant cells use this moment to set their degree of cytoplasmic continuity comes from a detailed analysis of early fern gametophyte development. Here, the densities of primary plasmodesmata formed during the first days of prothallium development gradually increase by an order of magnitude [41]. This might be especially important for fern development since no additional so-called secondary plasmodesmata (see Section 2.2) are formed. In line with this, the opposite phenomenon has also been observed in different fern tissues. In the roots of the floating fern *Azolla*, the density of the plasmodesmata formed by the root apical cell progressively drops by an order of magnitude as divisions progress, which could explain the eventual termination of root development [42]. 

The degree to which flowering plants rely on primary plasmodesmata is less clear. The high rates of cell division in, for example, the root apical meristem of higher plants, together with essential short-range signaling via plasmodesmata, suggest that primary plasmodesmata form an important gateway for early developmental patterning [43,44]. However, as cytokinesis-independent plasmodesmal biogenesis (discussed in the next section) is abundant, the relative contribution of primary plasmodesmata to overall connectivity is difficult to assess. 

### 2.2. Secondary Plasmodesmata and Diversification of Plasmodesmal Shape

Plasmodesmata can also be newly established independent of cytokinesis, including on cellular interfaces that are not directly clonally related. These plasmodesmata are called secondary plasmodesmata, and, broadly speaking, two major types can be discriminated: those templated by an existing plasmodesma and those inserted independent of a nearby plasmodesma (Figure 1B). 

The first type of secondary plasmodesma biogenesis is known as ‘twinning’, as the new plasmodesmal channel is inserted in close proximity (on the order of 100 nm) of an existing one (Figure 1B). Successive amplification in close vicinity leads to the grouping of plasmodesmata, and such plasmodesmal clusters are referred to as pit fields. Twinned plasmodesmata have been observed in diverse tissues across many higher plants, and their emergence appears to broadly coincide with later differentiation stages and cell growth [18,19,45]. The most comprehensive view of how plasmodesmal numbers and arrangements change via twinning comes from monitoring face-on views of the leaf epidermis–trichome cell wall segment [46]. The outer edges of this interface, where the increase in cell wall surface area is relatively high, showed the highest degree of plasmodesmal amplification [46]. The authors also found that sites of plasmodesmal amplification coincided with areas of the cell wall that were particularly rich in pectin and poor in cellulose [46,47]. Thus local cell wall growth rate and chemistry may feed back onto processes driving the secondary formation of plasmodesmata via twinning [48].

Next to twinning, plant tissues can build new plasmodesmata without the presence of pre-existing plasmodesmata. This type of secondary plasmodesmal biogenesis is called ‘de novo secondary plasmodesma formation’ (Figure 1B) [49]. It happens during many facets of normal higher plant development, including in the root meristem [44], the vascular cambium [50], and during leaf development [51]. Their formation is often required to establish developmentally relevant connectivity between non-clonally related cell layers, such as amongst the layers of the shoot apical meristem in higher plants (see later discussion on trafficking of transcriptional regulator LFY, Section 3.1). Comparative analyses showed that secondary plasmodesma formation correlated strictly with a meristem organization comprising multiple initial cells, suggesting that secondary plasmodesmal formation represents a formative element in this context [52].

The best-known signaling pathways that control secondary plasmodesmata formation come from a mutagenic screen for altered cell-to-cell trafficking in Arabidopsis embryos. Here, mutants showing enhanced intercellular trafficking were isolated and named increased size exclusion limit (ISE) mutants [53]. Subsequent functional analysis of the genes associated with the *ise1* and *ise2* mutants during leaf development showed that the absence of ISE1 and ISE2 increases plasmodesmal density, especially on cellular interfaces exclusively populated by secondary plasmodesmata, and enhances conductivity across these interfaces [51]. *ISE1* and *ISE2* encode, respectively, for a mitochondrial and plastidal RNA helicase [19,54,55]. Intriguingly, for both mutants, the downstream effects seem to impinge on plastid status: *ise1* and *ise2* loss of function both lead to less oxidized plastids [56], and the overlapping transcriptional changes in these mutants mostly relate to plastid status [49]. Indeed, widening the functional analysis to more molecular players of plastid physiology consistently showed that they affect plasmodesmal densities [57]. Whether cellular decisions on the construction of secondary plasmodesmata are always relayed via plastids or whether independent mechanisms exist remains to be elucidated. 

The less oxidized state of chloroplasts in the *ise1* and *2* mutants is mirrored by enhanced levels of plastid oxidants, leading to diminished intercellular trafficking [58]. However, this effect stems mainly from the occlusion of existing plasmodesmata rather than from a reduction in their frequency [58]. Thus, redox-derived signals can influence intercellular conductivity via both frequency effects and the control of the function of individual plasmodesmata. Other loss-of-function mutants for plastid function that are accompanied by increased cell-to-cell transport show both enhanced as well as reduced (similar to *ise* mutants) redox status in their plastids, showing that the coupling between a lowered plastidal redox status and enhanced cell-to-cell trafficking is not universal [55,59]. Overall, this hints at multiple, possibly even antagonistic, signaling pathways between chloroplasts and plasmodesmata, with plastid redox status being only one ingredient (reviewed in [60]).

Analyses of 2 other *ise* mutants (*ise3* and *ise4*) have revealed that nutritional signaling by the TARGET OF RAPAMYCIN (TOR) signaling network also controls plasmodesmal transport [61]. Impaired TOR signaling in *ise3* and *4* mutants increased intercellular trafficking via plasmodesmata. However, the contribution of secondary plasmodesmal biogenesis to this trait is not yet clear. 

While the signaling behind the de novo formation of secondary plasmodesmata is beginning to be untangled, comparatively little is understood about the cellular mechanisms that execute the actual process. Grafting has been used as an experimental platform to shed light on this, as grafting leads to a rather defined moment and place for the onset of secondary plasmodesma formation. In this setting, cell wall thinning coincides with secondary plasmodesmal biogenesis [62]. Then, as invaginations into the cell wall matrix form, a close association between the ER and the plasma membrane is consistently observed [63]. This could mean that ER-to-plasma membrane tethering activity is involved during de novo plasmodesma formation. With the molecular players of tethering during formation of primary plasmodesmata recently unearthed (see above, Section 2.1), it will be interesting to see if (the same) tethering proteins have overlapping roles during secondary plasmodesmal biogenesis.

The cellular machinery that drives the invasion of the plasma membrane and ER into the cell wall and the subsequent fusion with those membrane systems of the neighboring cell remains particularly elusive [19,62]. The occurrence of arch-shaped, plasmodesma-like structures that loop back from a cell’s exterior onto itself (also termed hemi-plasmodesmata), suggests that starting de novo plasmodesmal biogenesis does not strictly require a concerted action between two cells and that wall thickness might be a bottleneck to achieve a successful intercellular bridge [63,64]. With graft-induced secondary plasmodesmata formation available as an experimental system in Arabidopsis, the vast genetic resources of this model system could be leveraged to identify the processes driving de novo plasmodesmal biogenesis [64].

Besides the different ontologies, morphological variation of plasmodesmata also exists. Plasmodesmal channels can be simple in appearance, shaped as linear, cylindrical tunnels, and they can look like a networked tunnel system referred to as ‘branched’ or ‘complex’ plasmodesmata (Figure 1C). Additionally, on specific cellular interfaces important for symplastic post-phloem transport, funnel-shaped plasmodesmata are encountered (Figure 1C) [65]. In branched plasmodesmata, the central area located within the middle lamellae of the cell wall is often wider, forming a central cavity. Primary plasmodesmata appear to be exclusively laid down as simple plasmodesmata, with the potential to transition into complex plasmodesmata later in development. Newly formed secondary plasmodesmata can be formed as either morphological type, with the complex morphology assumed to be most typical (reviewed in [19,62]). The existence of plasmodesmata with a low degree of branching (i.e., having a Y- or X-shaped longitudinal profile), has been attributed to being an intermediary of twinned plasmodesmata or a more advanced stage that follows twinning via fusion (Figure 1C) [19,46]. Interestingly, the *ise1* and *ise2* mutants with higher plasmodesmal density discussed above also show higher numbers of branched plasmodesmata [54]. While this could indeed point toward branching as a by-product of enhanced insertion of secondary plasmodesmata, the processes disrupted in the mutants may also independently trigger plasmodesmal transformation. 

Given the more erratic shape of branched plasmodesmata, intuitively, a less controlled passage and thus enhanced SEL toward cargo molecules could be expected. However, the opposite situation appears to be mostly the case. In developing leaves, the bottom (proximal) part of the leaf acts as a sink tissue, while the distal tip has transitioned to a source tissue. In this context, a marked correlation was found between the capacity for intercellular trafficking of proteins up to 50 kDa and the proportion of simple versus complex plasmodesmata present, both being higher in sink tissues [66]. Likewise, the increase in the fraction of complex plasmodesmata along the developmental axis of Arabidopsis roots correlates with a lower overall degree of transport [44]. The transition to complex morphologies thus generally coincides with a lowered capacity for trafficking. 

It is currently hard to address the exact coupling between plasmodesmal ontology and morphology (e.g., is the transition of a simple plasmodesma to a complex plasmodesma different for those of primary or secondary origin?). The electron microscopy required to obtain the necessary resolving power to determine plasmodesmal ultrastructure makes the live tracking of plasmodesmal origins and life history impossible by definition. Therefore, live-cell, light-based fluorescence microscopy with higher resolving power in combination with markers that distinguish plasmodesmal morphology type could be a promising approach. The expanding knowledge of protein presence at different plasmodesmata via proteomics could provide candidates for such markers [67]. 

### 2.3. Dynamic Regulation of Plasmodesmata

Once plasmodesmata are generated, they do not exist as static nanoscopic tunnels between cells. Plasmodesmata are dynamically regulated to adjust intercellular flux according to tissue type, physiological parameters, and developmental stage. The mechanisms for this known to date include local cell wall modification at plasmodesmata and pressure-driven alterations to the membrane configuration. Also, the actin cytoskeleton has been implicated in controlling plasmodesmal function. In this section, these dynamic control mechanisms will be briefly addressed, and recent advances will be highlighted.

#### 2.3.1. Callose Turnover as Tuning Mechanism for Plasmodesmal Function

The local cell wall environment near plasmodesmata is different from that of the global cell wall, and this is relevant to plasmodesmal function. The cell wall polymer callose is highly enriched at plasmodesmata and forms a major component for cellular control over trafficking capacity (reviewed in [48,68,69,70]). Callose is mostly present in the neck regions of plasmodesmata, forming collars surrounding the plasmodesmal entrance [71]. Callose levels in this region set the level of constriction of the neck region [71,72]; Bringing the plasma membrane inward by callose deposition lowers the cross-sectional area of the cytosolic sleeve, thus reducing the SEL and the effective space available for intercellular diffusion (Figure 2A,B).

The amount of callose in the collar is set via the balance of two opposing enzymatic activities: the synthesis of polymeric callose from monomeric sugar precursors and the hydrolysis of the polymer. The enzymes responsible for callose synthesis and breakdown are, respectively, called callose synthase (CalS; also known as glucan synthase-like, GSL) and beta-1,3-glucanase. Several family members are typically encoded in a plant genome, and different members can have a tissue or context-specific role at plasmodesmata [68,72,73,74]. Several accessory proteins with regulatory functions have been identified that can integrate internal and external cues and relay them toward enzymes affecting callose homeostasis [68,75]. Such changes typically take place on the timescale of minutes. Together, this makes callose-induced alterations to plasmodesmal function highly dynamic and responsive to many inputs, including gibberellins, reactive oxygen species, and immune-related hormones [48,68].

Interestingly, in Arabidopsis MCTP tethering protein loss-of-function mutants, it was recently observed that in their roots, the stress-induced downregulation of intercellular connectivity could not be achieved despite the normal build-up of extra plasmodesmal callose [36]. Thus, for callose-induced neck constriction to properly downregulate the effective permeability of plasmodesmata in Arabidopsis roots, tethering of the desmotubule to the plasma membrane is required. How universal this requirement is and how, mechanistically, the interplay between local membrane and cell wall constellations works are exciting future avenues. 

Callose homeostasis at plasmodesmata is used by various plant tissues to regulate the proper dissemination of developmentally relevant cargoes. For instance, to form dormant shoot meristems in perennials at the onset of winter, this tissue becomes unresponsive to proliferation-promoting signals through the deposition of plasmodesmal callose [76,77]. Conversely, the controlled breakdown of callose in the following spring is a decisive factor for the meristem to activate [73]. Thus, the transient isolation of organs from developmental inputs can be achieved via callose homeostasis. A second example comes from stomatal development, during which a particular callose synthase (CalS10/GSL8) is critical for restricting plasmodesmal transport [78]. In the absence of this, cell-fate-specifying proteins inadvertently move away during the spatial specification of the stomatal lineage, leading to disrupted stomatal patterning in the epidermis [78]. 

Another developmental context in which callose at plasmodesmata determines the outcomes of intercellular signaling is the Arabidopsis root meristem. In gain-of-function mutants of callose synthase CalS3/GSL12 active in the root meristematic region, callose levels at plasmodesmata increase, leading to reduced mobility from phloem tissues [72]. To investigate the roles of symplastic communication in meristem organization, a hyperactive callose synthase allele was inducibly expressed in particular root tissues. This led to cell file specification defects and the loss of cell division coordination [43,72]. Some of these defects could be related to the impaired distribution of specific developmental regulators, including transcription factor SHORTROOT (SHR) [72] (see later in this review; Section 3.2). 

#### 2.3.2. Turgor Pressure and Plasmodesmal Conductivity

Turgor pressure is an osmosis-driven process and a hallmark of plant cells, vital to basic functions, such as growth and phloem transport [79,80]. Surprisingly, turgor pressure gradients occur not only between symplastically isolated regions but also in tissues with symplastic continuity [81,82]. Theoretically, a specific pool of osmolytes larger than the plasmodesmal SEL (or otherwise prevented from plasmodesmal transit) could lead to concentration differences between neighboring cells (e.g., by differential import, export, or turnover) and consequent differential turgor pressure. Furthermore, this would allow osmotic flow in a symplastic context [83]. These pressure differences could then even lead to an advective flow through the cytosolic sleeve. It will be interesting to learn if such scenarios truly occur *in planta*.

There are also indications for reverse causation, in which pressure differences between cells diminish plasmodesmal conductivity. For instance, changes in pressure between cells of the alga *Chara corallina* have been shown to result in less symplastic transport [84,85]. Furthermore, using microinjection in trichome cells of *Nicotiana clevelandii*, it was shown that symplastic transport between neighboring cells is strongly reduced upon a change in pressure [86]. A structural or physical mechanism operating at the plasmodesmal level underpinning these observations was, however, lacking. A recent modeling study gained insight into this by approaching plasmodesmata as a mechanosensitive system [87]. In this proposed system, under a pressure difference across a plasmodesma, the position of the desmotubule and associated ER becomes the product of two forces interacting with each other: one force stems from the pressure difference dislocating the desmotubule–ER complex, and, secondly, an elastic force resists this, originating from spoke-like elements residing in the cytosolic sleeve acting as springs [87] (Figure 2C,D). In this model, under increasing pressure, the space available for diffusion in the zone of cytosol at the approach toward the cytosolic sleeve becomes smaller. This effect is gradually exacerbated at larger pressure differences until, at a cut-off pressure, the path for symplastic flow becomes blocked [87].

For the spoke-like elements to act as springs that ‘memorize’ a basic position for the desmotubule as assumed in the model, they would need to have a very rigid interaction with the plasmodesmal membrane systems and a very low rate of turnover (at least compared with the timescale that the pressure difference would occur over). Key molecular candidates for constituting spokes-like elements with such properties are the earlier-discussed MCTP-type ER-to-plasma membrane tethering proteins, as they form immobile, high-molecular-weight complexes [26]. Although the biological relevance of the pressure differential-induced closure of plasmodesmata remains to be studied, a possible biological role could be that such a mechanism acts as a biomechanical safety valve; closing plasmodesmata upon rapid differences in turgor pressure, arising, for instance, from cell wall or membrane damage in one cell, would protect neighboring cells from sudden shifts in their cytosolic contents.

#### 2.3.3. Control by Actin and Associated Proteins

In plants, the actin cytoskeleton and associated motor proteins called myosins have roles in directional growth processes and organelle movement. Early immunological localization studies placed both these components at plasmodesmata [71,88] (reviewed in [89,90]), implying a function at plasmodesmata. In *Tradescantia* stamen hair cells, the pharmacological inhibition of myosin proteins was shown to differentially interfere with plasmodesma-mediated trafficking; drugs preventing myosin-to-actin attachment increased transport rates, while drugs reinforcing the attachment and blocking myosin movement decreased transport [89]. The same opposing effects have been observed in Lily [91]. However, preventing myosin-to-actin attachment in various monocot species led to the constriction of the plasmodesmal neck region, suggesting diminished transport capacity [71]. Myosin-based regulation of plasmodesmal conductivity may thus depend on the local plasmodesmal context, such as age or morphology [92]. Alternatively, secondary effects on plasmodesmata, originating from disrupting global myosin function, may have plant- or tissue-specific aspects, clouding the interpretation of the direct effects of the drugs on plasmodesma-associated myosins. 

Recently, membrane-bound members of a family of actin filament-nucleating proteins called formins [93] were found to reside at plasmodesmata [90]. In Arabidopsis, a lack of two closely related, plasmodesma-associated formins (FH1 and FH2) leads to increased plasmodesmal trafficking [94]. Disabling FH2′s ability to interact with actin mimicked this defect. Since FH1 is an actin nucleator, while FH2 unconventionally only caps and thereby stabilizes actin filaments [94], it remains an open question how the local actin configuration at plasmodesmata is disrupted in the formin mutants and how this, in turn, regulates intracellular trafficking. Regardless, this formin-mediated control over actin turnover at plasmodesmata, combined with the effects of myosin inhibition, suggests that actin cytoskeletal dynamics rather than actin presence per se is the most important for plasmodesmal regulation. In line with this, the depolymerization of actin has limited effects on plasmodesmal ultrastructure and functioning [25,89,90]. 

Where actin and associated proteins fit in the canonical topology of a plasmodesma is still uncertain. In earlier models of plasmodesmal structure, actin and myosin were seen as integral structural components surrounding the desmotubule. However, the extreme size constraints in the cytosolic sleeve, combined with a mismatch between actin/myosin sizes and the size of globular structures located in the cytosolic sleeve observed in transmission electron microscopy, indicate that this may not be a universal scenario (reviewed in [95]). Alternatively, a function for actin near the orifice of the plasmodesma could be envisaged. Since Arabidopsis NETWORKED 3C, a member of an actin-binding protein family, forms a higher-order complex with proteins generating ER-to-plasma membrane contact sites [96], actin near plasmodesmata may, for example, be involved in the shaping of the ER near the plasmodesmal entrance. 

## 3. Molecular Mechanisms Regulating Transport of Proteins Carrying Developmental Instructions

The mode by which plasmodesmal cargo proteins can end up in a neighboring cell can be coarsely divided into two methods of passage: a non-targeted (also called ‘passive’ or ‘non-selective’) and a targeted fashion (also described as ‘active’ transport). In this section, we address known mechanisms underlying both non-targeted and targeted movement of development-regulating proteins by discussing well-studied examples. 

### 3.1. Non-Targeted Transport 

Small proteins dissolved in the cytosol can traverse through plasmodesmata, irrelevant of whether they are native to the plant or not. This is well-illustrated by the ability of GFP (originating from jellyfish *Aequorea victoria*) to traverse through plasmodesmata with a sufficiently high SEL [97]. This mode of symplastic transport of proteins is known as non-targeted transport and is typically defined as intercellular protein movement without this cargo having interactions with the plasmodesmal apparatus beyond steric ones. The non-targeted movement of proteins is independent of specific domains within the protein sequence [97,98]. It is driven by diffusion along a concentration gradient or involves a type of advective flow due to hydrodynamic forces at the plasmodesma [97,99]. The hydrodynamic (Stokes) radius of a protein, derived from its size and charge, thus becomes the main determinant for the potential rate of cell-to-cell transmission via plasmodesmata [100]. In a tissue context, this means that the density and collective permeability of the plasmodesmata on a particular cell–cell interface, combined with a protein’s concentration gradient across this interface and its diffusive properties, together set the patterns of non-targeted transport (Figure 3A).

A key developmental regulatory protein exhibiting non-targeted movement is the transcription factor LEAFY (LFY) [101]. *LFY* is expressed in the floral meristem, where it regulates the transition into flowering [102]. The meristem of Arabidopsis is divided into three layers; *LFY* RNA is expressed in the outer L1 layer and is non-mobile. However, the LFY protein traverses to the deeper L2 and L3 layers, where it activates floral homeotic genes [101,103]. No protein domain required for its movement has been identified, nor has LFY been shown to be associated with any of the targeting mechanisms known for other mobile transcription factors discussed later in this review. Because of this, to date, it is assumed that LFY movement is purely non-targeted [16,102]. The flow of LFY within a tissue would, therefore, be solely dependent on the presence and conductivity of plasmodesmata within the shoot apical meristem (SAM). Each SAM layer is clonally distinct, meaning that all plasmodesmata between cell layers are secondary plasmodesmata and highlighting that the SEL of these secondary plasmodesmata is sufficiently high for trafficking a LFY–GFP fusion (74 kDa) [102].

### 3.2. Targeted Transport

Some proteins exceeding the baseline SEL or residing at intracellular locations making them not immediately available for transport can still be cell-to-cell mobile. Such proteins traverse the plasmodesmata in a targeted manner. In contrast to non-targeted transport, targeted transport relies, at least in part, on a specific interaction between cargo and the plasmodesmal apparatus [98,104]. Intriguingly, many developmental regulatory proteins that are trafficked from cell to cell are found to have a diverse set of targeting principles associated with them, which are surveyed in this section. We will explore several intracellular locations, modifications, and interactions that contribute to the targeted transport of proteins acting as developmental regulators by discussing examples for each known mechanism. Note that because of this, not all individually studied examples of targeted mobile proteins are covered.

#### 3.2.1. Nuclear Localization Required for Intercellular Protein Transport

Transcription factors form important nodes in the genetic networks that steer developmental processes and, interestingly, are often directly involved in cell-to-cell signaling. In fact, studies have extrapolated from surveying the overlap in expression domain and GFP-localization for a subset of transcription factors that in Arabidopsis roots, 25 to 29 percent of transcription factors are cell-to-cell mobile and that this mobility mostly involves targeted transport [105,106]. The first transcription factor found to move in a targeted fashion via plasmodesmata is the homeodomain (HD)-containing transcription factor KNOTTED1 (KN1) of maize [107,108]. In situ hybridization of KN1 mRNA on shoot apical meristems of maize shows expression in the L2 layer of the meristem. However, immunolocalization experiments reveal that KN1 protein is present in both the L2 and L1 layers [108], showing that the KN1 protein is mobile within the shoot apical meristem. Micro-injection of the protein alongside probes for intercellular connectivity in a heterologous system revealed that KN1 increases the SEL of plasmodesmata, characterizing the intercellular movement of KN1 as targeted. The HD region of KN1 is required to enable the intercellular trafficking of the protein [108]. Like KN1, its homologs in Arabidopsis, SHOOT MERISTEMLESS (STM) and ARABIDOPSIS KNOTTED-LIKE 1 (KNAT1)/BREVIPEDICELLUS (BP), are also mobile within the SAM, and this movement is similarly dependent on the HD motif [14,108,109]. 

The importance of the HD motif for intercellular transport was emphasized via a so-called trichome rescue assay [14]. In this assay, which has since become an established method to discover targeted mobile proteins, a suspected plasmodesmata trafficking competent protein (or domain) is fused to the otherwise non-mobile GLABROUS1 protein. If the protein confers mobility, the expression of this fusion construct from the sub-epidermal cell layer can restore trichome development in the epidermis of the Arabidopsis trichome deficient *glabra1* mutant. In the case of KN1, the N-terminal part of the HD motif contains a nuclear localization sequence (NLS), which was shown to be of key importance in enabling trafficking in the trichome rescue assay [14,109]. The need for a functional NLS sequence for the successful transportation of KN1 could indicate that nuclear residence plays a role in promoting the protein’s mobility. However, it remains unknown whether only the NLS domain from KN1 enables this mobility or whether any nuclear localization signal is sufficient for the targeted transport of KN1. 

The GRAS-type transcription factor SHR is expressed in the stele of the root in Arabidopsis, from which it moves via plasmodesmata to the quiescent center and the endodermis of the root apical meristem [110,111]. SHR is located in the cytoplasm and the nucleus of stele cells and is largely restricted to the nucleus in the endodermis. The capacity for intercellular movement of SHR correlates with its nuclear location [112]. Similar to KN1, SHR truncation mutants with impaired nuclear presence also exhibited a lack of intercellular movement. However, the fusion of a nuclear-localized version of GFP to SHR, leading to nuclear hyperaccumulation, prevented its movement from the stele, showing that nuclear retainment can act as a negative factor [98]. In line with this, SHR’s molecular interactions in the nuclei of the endodermis are understood to diminish further SHR transport (Figure 3B) [113,114]. To determine whether the nuclear localization of SHR is required for intercellular mobility, a mutant *SHR* allele encoding a single T > I point mutation was investigated. This mutant version of SHR exhibits poor nuclear import accompanied by a lack of intercellular transport. By the fusion of an additional, unrelated NLS to this variant, both nuclear import and cell-to-cell transmission could be restored. This shows that the mutation does not directly cause a lack of intercellular transport and supports the notion that nuclear presence of SHR is relevant to the mobility trait [112]. Taken together, this suggests that SHR cell-to-cell movement requires a transient localization to the nucleus, where it is possibly altered, followed by further cytosolic processing for transport (Figure 3B).

Intercellular movement of TARGET OF MONOPTEROS 7 (TMO7) in Arabidopsis also depends on both nuclear import and export. This transcription factor controls the identity and cell divisions in the hypophysis of the embryo where it enters from the upper embryonic cells. It was also shown to be mobile within seedling roots, where it can move from meristematic and lateral root cap cells into columella cells and the quiescent center in a targeted manner [115,116]. Experimentally imposing the confinement of TMO7 localization within the nucleus as well as its forced exclusion abolished protein movement, showing that transient nuclear localization is part of a successful intercellular movement pathway for TMO7 [115]. The observation that the disruption of putative phosphorylation sites in TMO7 lowers its transportability but does not affect its subcellular localization suggests that post-translational modifications are involved [115].

The sequence driving nuclear import of the MYB-type transcription factor CAPRICE (CPC), which plays a role in the alternating patterning of hair and non-hair cells in the root epidermis [117], has also been described to influence its intercellular movement. The mobility of CPC was found to depend on its N-terminus and the MYB domain. The amino acid W76 located in the MYB domain was found to be critical for cell-to-cell movement as well as the nuclear localization of CPC. This region of the protein does not resemble any known nuclear localization signal, suggesting an unknown type of nuclear import [118]. However, the fusion of the CPC N-terminus and its MYB domain to GFP was not sufficient to enable the cell-to-cell transport of the fusion protein. This indicates that either these two domains alone are not sufficient to confer mobility or that the three-dimensional structure of CPC is ultimately important for its mobility. Furthermore, CPC was found to be mobile in the epidermis but not in the stele, suggesting the involvement of tissue-specific regulators or differences in plasmodesma structure between these cell types [119].

#### 3.2.2. Role of Endosomes and the Cytoskeleton in Intercellular Transport

In addition to the nucleus, endosomes have emerged as a subcellular location where cargo transportability can be regulated. The involvement of endosomes stems from the discovery of interaction between SHR and the SHORTROOT INTERACTING EMBRYONIC LETHAL (SIEL) protein that resides at endosomes [120]. Endosomes form a sorting hub for endo- and exocytotic trafficking, and SHR localizes broadly to several endosomal subtypes in an SIEL-dependent manner [121]. The pharmacological disruption of the formation of particularly the early and late endosomes, but not the recycling type (i.e., those directed from the endomembrane compartment toward the plasma membrane), correlated with diminished SHR intercellular trafficking [121]. According to these results, SIEL was proposed to bridge SHR to an endosome-bound activity that ultimately promotes its intercellular transit (Figure 3C). 

The model of SIEL-promoted targeted transport of SHR was expanded with the discovery that the disruption of microtubules interferes with SHR intercellular movement and the identification of a plant-specific kinesin (KinG) that interacts with SIEL [122,123]. KinG is a calponin homology domain kinesin, which localizes primarily to the microtubule cytoskeleton and additionally shows enrichment at actin and static foci in the ER, indicating that it can reside at the crossroads between these structures [122,124]. Live cell imaging in a heterologous tobacco system indicated that KinG is a nonmotile kinesin that promotes the pausing of SHR-associated endosomes. The mechanism through which pausing then leads to the promotion of SHR transport is currently unknown. It might involve the assembly of a movement-competent protein complex or protein modification to promote the movement of SHR through plasmodesmata [122]. The pausing suggests that motility of the SHR-associated endosomes themselves is unlikely to foster intercellular transport. This is further underscored by the fact that actin disruption only minimally diminishes SHR transport [123]. This probably makes the endosome-bound mechanism operating on SHR mobility distinct from that operating on viral replication complexes, where endosome motility correlates with enhanced localization at and transmission via plasmodesmata [125]. Intriguingly, besides SHR, SIEL also interacts with a range of other mobile transcription factors, including AGAMOUS-LIKE21 (AGL21), TMO7, and CPC, but not with LEAFY or STM [16,120]. This suggests that SIEL activity at endosomes extends to fostering the transport of multiple plasmodesmal cargo proteins. That this is functionally relevant for at least CPC transport is suggested by mutant SIEL plants phenocopying the *cpc* root hair patterning defects [120]. 

Interestingly, beyond its role at endosomes, SIEL also functions inside the nucleus [126]. SIEL was found to function in a small nuclear RNA 3′-end processing complex important for the splicing machinery [126]. Whether this pool of SIEL proteins is relevant to the regulation of SHR transport is not yet clear, but it may offer an intriguing link to the requirement of nuclear shuttling for several of the mobile transcription factors discussed in Section 3.2.1 (Figure 3B,C).

Retrograde endosomal trafficking of Arabidopsis KN1 homolog STM away from the plasma membrane has been implicated in suppressing its intercellular mobility. The coupling of STM to endosomes was found to depend on the tethering proteins MCTP3 and MCTP4. In loss-of-function mutants of MCTP3 and MCTP4, enhanced STM presence at the cell periphery was observed, combined with enhanced intercellular mobility [127]. This suggests that beyond functioning directly at plasmodesmata (see Section 2.1 and Section 3.3), MCTPs may have additional functions at endosomal membrane compartments to control intercellular movement. However, possible effects on baseline plasmodesmal function in the MCTP3 and MCTP4 mutants cannot be ruled out. In fact, in root tissues, these mutants exhibit enhanced plasmodesmal conductivity [36], while in leaf tissue, diminished conductivity has been observed [35]. Thus, establishing the baseline conductivity in SAM tissues is important for gauging the precise contribution of the proposed endosome-mediated pathway for STM intercellular mobility. 

In the case of KN1, the microtubule-associated protein MPB2C was found to negatively regulate the intercellular movement of KN1 by binding to its HD motif [128]. The authors in [128] speculated that MPB2C may be part of a microtubule-associated complex with chaperone-like functions. In their model, only correctly folded or modified KN1 protein bound to its own mRNA was allowed to move through plasmodesmata, escaping MBP2C binding. MPB2C-mediated retention of misfolded or non-functional KN1 at microtubules would instead lead to its rapid degradation (Figure 3C,D). Thus, alongside providing regulatory functions by interacting with endomembrane compartments as seen with SHR, the microtubule cytoskeleton could additionally regulate plasmodesmal cargo transport by providing a scaffold for biochemical pathways that modify the cargo.

### 3.3. How Does a Protein Traverse through Plasmodesmata?

Once a protein competent for movement has reached a plasmodesma, several molecular mechanisms could operate to facilitate its traversal through the cytoplasmic sleeve. If proteins are small enough to overcome the SEL of plasmodesmata in a particular cell type without interaction with components in the cytoplasmic sleeve, diffusion or advective flow would be sufficient. 

However, for targeted transport, additional mechanisms are likely at play. Two of these mechanisms are encountered during the passage of the KN1 protein. First, this protein can enlarge the SEL of plasmodesmata [129]. Second, KN1 translocation through the symplastic tunnel of the plasmodesma is facilitated by manipulating its tertiary structure [129]. Microinjection studies with KN1 proteins, which were fixed in their tertiary structure, suggested that KN1 needs to be (partially) unfolded to allow passage through plasmodesmata [130]. This theory was experimentally confirmed by trichome rescue assays, showing that KN1 passes plasmodesmata in an unfolded state and is re-folded into a functioning protein once it has reached the target cell [131]. This re-folding of KN1 in the target cell is facilitated by the CHAPERONIN CONTAINING T-COMPLEX POLYPEPTIDE 1 (CCT) 8, a subunit of the type II chaperonin protein complex (Figure 3D). The reduced expression of other chaperonins that are part of this complex also resulted in the loss of intercellular trafficking of KN1 in the trichome rescue assay. This suggests that not only the subunit CCT8 but also the whole chaperonin complex is involved in the intercellular trafficking of KN1 [131,132]. The successful transport of KN1 in the SAM is hampered in *cct8* mutants, and CCT8 is also required for the successful intercellular movement of the Arabidopsis KN1 homolog STM and the structurally unrelated TRANSPARANT TESTA GLABROUS (TTG1) protein, a protein involved in regulating trichome spacing. However, CCT8 is not required for the movement of SHR, indicating that this unfolding and refolding mechanism is not universal for targeted intercellular transport [131]. 

Another group of proteins involved in regulating traversal through plasmodesmata are the previously introduced MCTPs. Apart from providing structural interconnections between plasmodesmal membranes (see Section 2.1), MCTPs can also affect the mobility of specific cargoes by interacting with the cargo. For example, Arabidopsis FTIP1 (Flowering Locus T Interacting Protein 1, also termed MCTP1) interacts with the mobile florigen Flowering Locus T (FT), enabling its cell-to-cell transport (Figure 3E) [133]. FT in Arabidopsis and its homologs in other flowering species are famous regulators of the transition to flowering. FT is produced in the leaf vasculature and transported via the phloem to the SAM, where it promotes the expression of several flowering-inducing genes. To be loaded into the phloem, FT must move from companion cells into the sieve elements, where FTIP1 is localized at plasmodesmata [134]. The loss of function of FTIP1 results in the obstructed export of FT from companion cells, suggesting that the FT–FTIP1 interaction in these cells is critical to successfully guiding FT into sieve elements [133,134]. Other features of the FT protein also play a role in determining its mobility [135]. FT interacts only with the third C2 domain of FTIP1, but how this binding enables its traversal through the cytoplasmic sleeve remains to be elucidated [133]. FT transport is also dependent on another MCTP known as QUIRKY (QKY; MCTP15) via an interaction with the plasma membrane-localized t-SNARE SYNTAXIN121 protein. In contrast to FT’s interaction with FTIP1 at plasmodesmata, its interaction with QKY predominantly takes place at endosomes [136]. Thus, endosomal and plasmodesmal MCTP-mediated pathways may work in parallel to control the cell-to-cell movement of FT.

The involvement of MCTPs in controlling plasmodesmata-mediated transport does not only entail FT and its homologs. CPC movement was found to be regulated by a leucine-rich repeat containing receptor-like kinase called SCRAMBLED (SCM, also known as STRUBBELIG) and its interacting MTCP protein QKY. Complex formation between QKY and SCM was found to stabilize SCM at plasmodesmata by preventing SCM ubiquitination and thereby degradation in the vacuole. The accumulated SCM at plasmodesmata facilitates the import of CPC into the receiving cell [137]. Considering that SCM kinase activity is not required for epidermal cell patterning [138,139], it remains to be determined how SCM controls CPC import. 

## 4. Concluding Remarks

In this review, we aimed to present an overview of the role of plasmodesmal biology as well as modifications of their cargo proteins in the transport of transcriptional regulators in a developmental context. While we presented the plasmodesma-associated and cargo-associated mechanisms as co-existing, regulation on these different planes is likely to intersect. Such interdependence (e.g., to what extent are the different targeted transport mechanisms independent of SEL regulation?) is, however, even less well understood than the individual mechanisms. Also, the hierarchy in which plasmodesma-associated and cargo-specific mechanisms operate is seemingly diverse. There are examples from individual mobile protein cargoes that either of them can take precedence over the other. For instance, KN1 affects the plasmodesmal SEL and ‘overrides’ it, even in non-native contexts [108]. On the other hand, SHR trafficking is strongly affected by the modification of callose levels at plasmodesmata [72], highlighting that despite the many regulatory layers at the cargo level (Figure 3B,C), the dimension of the cytosolic sleeve still forms a dominant control node for the dissemination of this protein. Studying targeted mobile proteins in different plasmodesmal contexts can help to disentangle the possible hierarchies and synergies among the different regulatory mechanisms operating on plasmodesmata and their proteinaceous cargoes.

From the literature, it seems that most mobile transcriptional regulators appear to move between cells in a targeted fashion rather than in an untargeted fashion. This raises the question of whether targeted movement is the predominant mode of regulated protein transport in plants at large. While this observation could be due to an unknown bias in investigative strategies leading to a higher detection rate of targeted protein movement, it is impossible to ignore the multitude of cellular mechanisms setting the transmissibility of individual proteins (Figure 3). This might point to targeted movement providing a more robust way to control the intercellular distribution of proteinaceous cargos than non-targeted transport. The tuning of plasmodesmal conductance as a primary control mechanism likely has limitations. There are many plant physiological processes that rely on properly tuned plasmodesmal conductivity, such as metabolite flux and the mounting of immune responses. There could thus be conflicting needs when setting plasmodesmal conductance and/or SEL and steering for the optimized execution of one supracellular process might come at the expense of another. Orchestrating protein mobility regulation on an individual cargo basis might then be a way to alleviate such conflicts. These potential downsides could have led to the apparently wide adoption of targeted transport mechanisms for developmental regulatory proteins.

Interestingly, the large selection of targeted mobile proteins has hitherto not led to the identification of a universal ‘address label’ for plasmodesmal transit. If such a plasmodesmal sorting signal exists at all, it might not be encoded by the primary protein sequence but rather via a feature in the secondary structure or via post-translational modification. The absence of a unifying element in targeted protein transit regulation so far thus favors a view of several distinct regulatory mechanisms that act in parallel. However, for most targeted mobile transcription factors, the element at plasmodesmata that recognizes and facilitates the actual transit of their movement competent form is still unknown, and this might form a missing link to a more centralized mechanism that selects for protein intercellular movement. The finding that effector proteins translocated by a fungal pathogen into the host plant cytosol also show hallmarks of targeted intercellular transport [140] expands the catalog of targeted mobile proteins and opens up possibilities for mechanistic comparisons with native targeted proteins.

Since plant development is chiefly driven by meristems, these form the tissue context for many mobile transcriptional regulators. Performing functional studies on plasmodesmal transport can be challenging in meristems given their relative experimental inaccessibility; they are typically embedded in deeper tissue layers and comprise a complex three-dimensional setting. The fundamental study of plasmodesmal transport mechanisms would thus benefit from the use of simple and experimentally tractable meristematic tissues. For this, land plants with simpler body plans, such as bryophytes, could prove highly useful. Their meristems have a simpler organization while at the same time still sharing genetic control elements with higher plants [141]. Furthermore, knowledge of the molecular makeup of bryophyte plasmodesmata is emerging, along with tools for studying plasmodesmal function [67,142,143]. Beyond the advantages in terms of a more accessible body plan, the inclusion of bryophytes could shed an evo-devo light on the control mechanisms for mobile developmental regulators.

## Figures and Tables

**Figure 1 plants-13-00684-f001:**
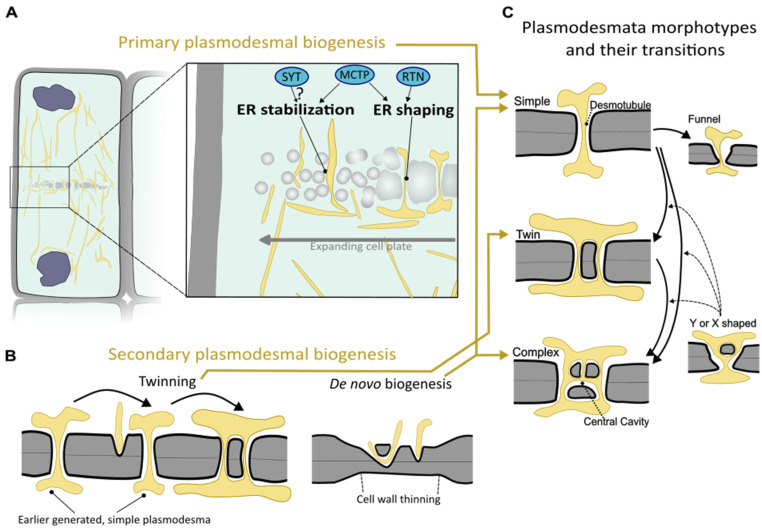
Schematic depiction of biogenesis of plasmodesmata and plasmodesmata morphotypes. (**A**) Primary plasmodesmata are synthesized during cytokinesis. A portion of the newly forming cell wall segment termed the cell plate in a dividing cell (**left**) is shown enlarged on the right. Here, strands of ER traverse the expanding cell plate to ultimately mature into plasmodesmata. In this process, ER strands bridging the expanding cell plate are stabilized by MCTP tethering factors and potentially synaptotagmins (SYTs). ER strands are shaped into the desmotubule, likely by membrane constriction via MCTP and reticulon (RTN)-type proteins. (**B**) Two means of secondary plasmodesmal biogenesis can be distinguished: a new plasmodesmal channel can be inserted in close proximity to an already existing plasmodesma through a process called ‘twinning’ (**left**). Furthermore, de novo biogenesis creates plasmodesmata via membrane penetration into an often thinned cell wall segment independent from an existing plasmodesma (**right**). (**C**) Different plasmodesma morphotypes and their possible transitions. Simple plasmodesmata can originate from either primary or de novo biogenesis. Simple plasmodesmata can become twinned plasmodesmata via twinning (see (**B**)). On highly specialized intercellular interfaces, simple plasmodesmata can attain specialized morphologies, such as a funnel-shaped morphology. Complex plasmodesmata are characterized by a central cavity and a branched desmotubule and can arise through the modification of simple or twinned plasmodesmata, but they can also arise during de novo secondary biogenesis. X- or Y-shaped plasmodesmata have an unresolved relationship with the various transitions (being intermediaries during twinning or transformation toward a complex morphotype; dashed arrows). Yellow arrows among (**A**–**C**) point to the plasmodesmal morphologies that the biogenic processes can give rise to.

**Figure 2 plants-13-00684-f002:**
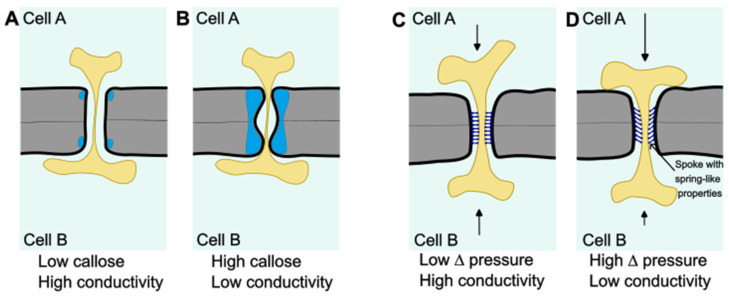
Modulation of plasmodesmata conductivity by local cell wall modification and a proposed mechanism for pressure-induced plasmodesmata conductivity regulation. (**A**) When callose (indicated in blue) levels in the cell wall at plasmodesmata are low, the neck region of the symplastic tunnel is open, resulting in a relatively high SEL. (**B**) When callose levels at plasmodesmata are high, especially at the plasmodesmal entrance, the neck region is constricted resulting in a smaller SEL and less conductivity. (**C**) Schematic representing plasmodesmata between two cells with a low difference in pressure. Spokes-like elements (possibly formed by tethering proteins) keep the desmotubule in place. (**D**) When pressures between two cells differ strongly, the ER at the plasmodesma on the side of the cell with the higher pressure is pressed against the plasma membrane, causing the entrance of the cytoplasmic sleeve to close.

**Figure 3 plants-13-00684-f003:**
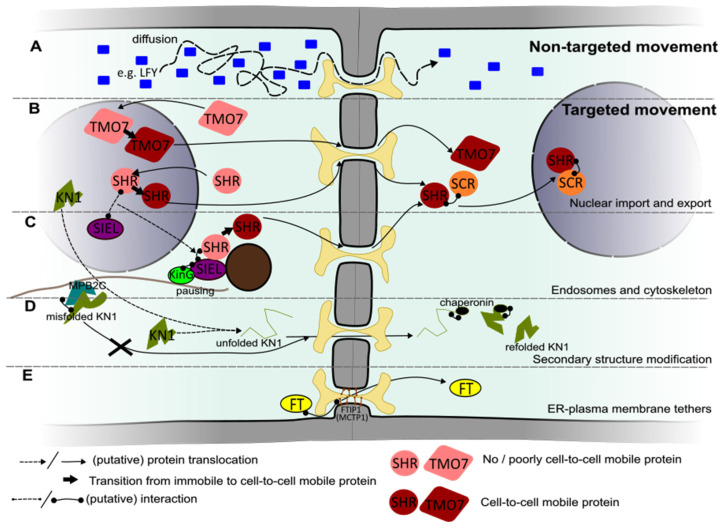
Examples of cargo-dependent regulatory mechanisms that control intercellular transport capacity of proteins carrying developmental instructions. (**A**) Non-targeted proteins (depicted by blue rectangles) like LFY travel from cell to cell via diffusion (and possibly advection). The rate of intercellular movement of these proteins is dependent on the dimensions of the cytosolic sleeve, the diffusive properties of the protein, and its concentration gradient across a cellular interface, but it does not involve active modification of either the cargo or plasmodesmal apparatus (processes described in (**B**–**E**)). Note that the translocation steps depicted in (**B**–**E**) may still be diffusion-driven. (**B**) The transcription factors TMO7 and SHR require a transient nuclear localization to become mobile from cell to cell. The processes occurring in the nucleus that enable the mobility of these proteins are unknown. SHR is retained in the nucleus of the neighboring cell by interaction with SCR, preventing SHR from moving to other cells. (**C**) Involvement of endosomes and the cytoskeleton in enabling intercellular protein movement. SHR localizes to endosomes via interaction with SIEL, enabling intercellular movement of SHR. The SIEL and SHR association is understood to be functionally relevant at endosomes, but it could possibly already happen in the nucleus (depicted in (**B**)) or the cytosol. SHR-associated endosomes (brown) are paused from moving by the microtubule cytoskeleton-associated KinG, during which an unknown mechanism promotes the intercellular movement capacity of SHR. Additionally, another cytoskeleton-based activity is involved in regulating KN1 protein transportability. The microtubule-associated MPB2C binds misfolded KN1 proteins, possibly preventing these proteins from moving to neighboring cells. (**D**) Chaperonins act in a receiving cell to refold proteins that traverse plasmodesmata in an unfolded state, such as KN1. (**E**) Members of the MCTP family tether the ER and plasma membrane at plasmodesmata. FT is a transcriptional regulator that physically interacts with FTIP1 (MCTP1), and this interaction fosters FT intercellular mobility via an unknown molecular mechanism.

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
