# Peer review of "Distributing Plant Developmental Regulatory Proteins via Plasmodesmata"

_plants, 2024, doi:10.3390/plants13050684_

Round 1

Reviewer 1 Report

Comments and Suggestions for Authors

The current review manuscript has comprehensively covered the details on the biogenesis, basic properties and function of plasmodesmata (PD) during various developmental processes of plants. Particularly, proteins that are regulating the dynamics of PD and proteins that are moved via PD are summarised. It would be of great values to readers who are working on or interested in PD and plant development. However, there are a couple of issues that need to be addressed.

 Major comments:

1. I found the manuscript is not easy to follow probably due to the mixed information of morphogenesis and regulation in each session, rather long sub-sessions, lack of informative figures as well as language in part. Perhaps the authors could consider to re-organize and shorten certain sessions to increase the fluenence of the article.

 2. The title of current version is focused only on the mobile proteins that could move via PD during plant development, however, almost half of the space was focused on the proteins that are involved in the biogenesis and modification of PD. Maybe the authors could re-think on the title.

 3. Another issue is that the authors discussed many molecular factors in the text that are related to the topics of Figure 1-3. Unfortunately, not any of them appeared in the figures. I would say current Figure 1-3 is not informative and I suggest add the most important moleculars to corresponding figure. Alternatively, a table of proteins/molecular factors could be generated for summary.

 4. In my opinion, “3.2.3. How Does a Protein Traverse through Plasmodesmata?” is not in parellel with 3.2.1 and 3.2.2, perhaps it could be 3.3?

 5. Involvement of organelles was highlighted in the abstract, but not many details were discussed in the text; similarly, cell-to-cell movement of phytohormone via PD was mentioned earlier in the manuscript, however, no further discussion was made on this aspect.

Minor comments:

-          I noted that the authors included many unconfirmed or uncertain mechanisms throughout the manuscript, I feel some of them are not necessary.

-          Figure 1, the biogenesis of PD is a dynamic process, however, Fig.1 is not demonstrating such process. Also, figure legens is not clear.

-          Figure 4, light red and dark red block are referred to “No/poorly cell-to-cell mobile protein” and “Cell-to-cell mobile protein”for TMO7, respectively. Does such colour code also apply to SHR? How is the “red” SHR different from dark red one?Actually, not all colours are annotated. Additionally, the figure legend could be more concise.

-          For the development of plasmodesmata, I wounder some marker genes such as PDCB1 and PDBG2 could be mentioned.

-          It would be interesting to show the tissue or cell type specificity of certain secondary PD in plants.

-          line 48, it would be great to describe the range of the aperture of PD in certain conditions and this is also linked to the non-target or target transport in the session 3.

-          Please specify all plant species mentioned in the text, for example, the perennials in line 364

-          Please check the use of abbreviation of gene names are correct through the text.

Comments on the Quality of English Language

I would suggest that the language could be revised to increase the fluency and readability of the manuscript, please reduce the frequency on using long and complicated sentences in the text.

Reviewer 2 Report

Comments and Suggestions for Authors

The review "Cellular control over mobility of proteins encoding plant developmental signals via plasmodesmata" gives a very good overview about the recent research regarding intercellular transport of proteins via plasmodesmata. It is written in a very comprehensive style and the bibliography contains very recent and significant publications in the field. Teh Bibliography is very substantial and almost complete. Current methods and model organisms are discussed and presented in detail.

Minor points:

It would be helpful to have an overview about the structure of the article or a table of contents right at the beginning.

Paragraphs about molecular mechanisms regulating transport of proteins 3.2.1 and 3.2.2 could be shortened; both paragraphs are much too long.

Sometimes protein transport via plasmodesmata can not be analyzed without knowledge about the transport capacity of the appropriate mRNA. Why is the transport of RNA molecules is not mentioned at all?

Reviewer 3 Report

Comments and Suggestions for Authors

This review focuses on the cell-to-cell transport of plant regulatory proteins through plasmodesmata. It is based on a large amount of experimental data and provides a reasonable overview of the relevant pathways and mechanisms. Therefore, in my opinion, the paper is worth publishing. However, I have some concerns that should be addressed before the paper can be accepted for publication.

Major points.

1. The introductory part on the structure and biogenesis of plasmodesmata (Section 2) is too long and even larger than the main part of the review (Section 3). Therefore, Section 2 should be significantly shortened.

2. The title: The wording should be changed as proteins do not encode signals. Consider 'Cellular control of plasmodesmata mobility of proteins serving as plant developmental signals'.

3. Proofreading by a native English speaker is mandatory.

Minor points.

1. There are some terminology problems in the text. The term 'cargo' is used incorrectly as it refers to objects that are carried, and it cannot be used in relation to proteins transported through plasmodesmata since the carriers of these proteins are not known and likely do not exist. Additionally, 'plasmodesmata' is always plural and should be used without an article.

2. Section 2.3.3 could benefit from mentioning that myosins VIII are involved in the localization of viral movement proteins to plasmodesmata and likely facilitate their transport through plasmodesmata (PMID: 18199648, 25329993).

3. Lines 130-131. Reticulons are not transmembrane proteins because they do not cross the lipid bilayer.

Comments on the Quality of English Language

Proofreading by a native English speaker is mandatory.
